

# The Biogeophysical Effects of Carbon Fertilization of the Terrestrial Biosphere

Robert J. Allen[1*]

[1]Department of Earth and Planetary Sciences, University of California, Riverside USA

[*]*Correspondence to*: R. J. Allen (rjallen@ucr.edu)

## Abstract

The response of the terrestrial biosphere to increasing atmospheric carbon dioxide ($CO_2$), i.e., the carbon fertilization effect represents a significant source of uncertainty in future climate projections. The climate impacts of carbon fertilization include cooling associated with the biogeochemical effects of enhanced land carbon storage, whereas the non-carbon cycle biogeophysical effects associated with changes in surface energy and turbulent heat fluxes may warm or cool the climate system. Here, I analyze 15 state-of-the-art Earth system models that conducted simulations driven by 1% per year increases in atmospheric $CO_2$ concentration that isolate the $CO_2$ fertilization effect (i.e., $CO_2$ radiative effects are not active). At the time of $CO_2$ quadrupling, the biogeophysical effects yield multimodel global mean near-surface warming of $0.16 \pm 0.09$ K with 13 of the 15 models yielding warming. Most of this warming is associated with decreases in surface latent heat flux associated with reduced canopy transpiration. Decreases in surface albedo and increases in downwelling shortwave and longwave radiation—both of which are modulated by cloud reductions—are also associated with the warming. Overall, however, the biogeophysical warming is about an order of magnitude smaller than the corresponding cooling associated with enhanced land carbon storage at $-1.38$ K ($-1.92$ to $-0.84$ K).

## Short Summary

CMIP6 models are analyzed to quantify the biogeophysical (non-carbon cycle) and biogeochemical (enhanced carbon storage) effects of carbon fertilization at the time of $CO_2$ quadrupling. The biogeophysical effects lead to relatively weak warming ($0.16 \pm 0.09$ K) largely due to decreases in surface latent heat flux associated with reduced canopy transpiration. Biogeochemical cooling associated with enhanced land carbon storage dominates at $-1.38$ K ($-1.92$ to $-0.84$ K).



## 1. Introduction

Over land, increasing atmospheric $CO_2$ concentrations are associated the carbon fertilization effect (e.g., Ainsworth and Long, 2005; Ainsworth and Rogers, 2007; Leakey et al. 2009; Norby and Zak, 2011).   This effect involves physiological and structural vegetation changes including reduced stomatal conductance and increased photosynthesis rates which are expected to increase net primary productivity (NPP) and carbon storage.  However, the carbon fertilization effect depends on many factors, including plant species, temperature, and availability of water and nutrients.  The availability of soil inorganic nitrogen (N), for example, exerts a strong control on plant productivity and carbon storage in many temperate and boreal ecosystems (Vitousek and Howarth, 1991; Oren et al., 2001; Fernández-Martínez et al., 2014; Kicklighter et al., 2019). Nonetheless, intensification of terrestrial biospheric activity, including increased global photosynthesis and "greening" of the planet, has been found in several recent studies (Forkel et al., 2016; Thomas et al., 2016; Zhu et al., 2016, Campbell et al., 2017; Keeling et al., 2017; Haverd et al., 2020; Walker et al., 2021; Chen et al., 2022; Keenan et al., 2023).

Enhanced land carbon storage and greening of the terrestrial biosphere under elevated atmospheric $CO_2$ concentrations will promote a biogeochemical cooling effect. In other words, the carbon-concentration feedback, which quantifies the carbon cycle's response to changes in atmospheric $CO_2$ concentration (expressed in units of carbon uptake/release per unit change in atmospheric $CO_2$ concentration) is negative from the atmosphere's perspective (Arora et al., 2020).  Such changes in the terrestrial biosphere will also drive biogeophysical effects associated with surface energy and turbulent heat fluxes.  For example, structural vegetation changes (e.g., enhanced leaf area index, LAI) associated with carbon fertilization will impact surface physical properties.  This includes altered surface albedo, e.g., plants are darker than bare soil (Betts et al., 2000, Bala et al., 2006; Li et al., 2015), which promotes enhanced surface absorption of solar radiation and hence warming.  Furthermore, the physiological changes of carbon fertilization (i.e., reduced stomatal conductance and enhanced water use efficiency) are associated which reduced plant transpiration and latent heat flux, which directly impact surface temperature (i.e., less evapotranspiration implies surface warming) as well as atmospheric water vapor and clouds (Field et al., 1995; Bounoua et al., 1999; Cao et al., 2010; Doutriaux-Boucher et al., 2009). Thus, the overall impact of carbon fertilization on surface temperature is determined by a combination of biogeochemical (carbon cycle) effects and biogeophyscial (non-carbon cycle) effects.

In this paper, I use 15 Coupled Model Intercomparison Project phase 6 (CMIP6; Eyring et al., 2016) models to quantify the climate effects of carbon fertilization, i.e., in the absence of the direct radiative effects of $CO_2$.  Climate effects include the directly simulated biogeophysical (non-carbon cycle) temperature response as well as the drivers, while I infer the biogeochemical temperature response. I find significant global mean biogeophysical warming, largely driven by reductions in latent heat flux associated with decreases in canopy transpiration.  Decreases in surface albedo and increases in downwelling shortwave and longwave radiation (which are modulated by cloud reductions) are also associated with the warming.  The magnitude of this biogeophysical warming, however, is about an order of magnitude smaller than the inferred biogeochemical cooling associated with enhanced land carbon storage.



## 2. Methods

### 2.1 CMIP6 Models and 1% per Year Simulations

CMIP6 (Eyring et al., 2016) performed three sets of 1% per year increasing atmospheric $CO_2$ concentration simulations (1PCTCO2), which are initialized from the preindustrial $CO_2$ concentration of ~284 ppm and integrated for 150 years. The default 1PCTCO2 simulations are fully coupled as the radiation and carbon cycle components see the increasing $CO_2$ concentration. Two variants of the 1PCTCO2 simulation were performed as part of the Coupled Climate-Carbon Cycle Model Intercomparison Project (C4MIP; Jones et al., 2016), including a biogeochemically coupled version (1PCTCO2-bgc) and a radiatively coupled version (1PCTCO2-rad). Under 1PCTCO2-bgc, only the carbon cycle components (both land and ocean) respond to the increase in $CO_2$, while the atmospheric radiative transfer calculations use a $CO_2$ concentration that remains at the preindustrial concentration. Under 1PCTCO2-rad, only the atmospheric radiation code sees the increase in $CO_2$ and the carbon cycle components see the fixed, pre-industrial $CO_2$ concentration.

The focus of this analysis is on the 1PCTCO2-bgc runs, which allows assessment of the climate responses associated with the carbon cycle under elevated $CO_2$ (without the influence of $CO_2$ radiative effects). Over land, this effect is traditionally referred to as carbon fertilization of the terrestrial biosphere (e.g., Ainsworth and Long, 2005; Ainsworth and Rogers, 2007; Forkel et al., 2016; Thomas et al., 2016; Zhu et al., 2016, Campbell et al., 2017; Chen et al., 2022). In particular, this includes changes in vegetation physiology including photosynthesis, transpiration and stomatal conductance, and changes in vegetation state (e.g., leaf area index, canopy height). In models with dynamic vegetation, this also includes changes in vegetation type and coverage. These changes in turn affect surface radiative and turbulent heat fluxes which impact surface temperature and other aspects of climate. As the three sets of 1PCTCO2 simulations are $CO_2$ concentration driven (as opposed to emissions driven), the simulated climate responses include only the biogeophysical effects (e.g., changes in surface fluxes and more generally all non-carbon cycle effects). The climate impacts associated with changes in terrestrial carbon pools (biogeochemical effects) are not allowed to feedback onto the climate system (i.e., enhanced land carbon storage under elevated $CO_2$ does not impact the atmospheric $CO_2$ concentration and thus does not impact climate). However, as discussed below, the surface temperature responses to changes in terrestrial carbon pools can be inferred from the transient climate response to cumulative $CO_2$ emissions (TCRE; Gillett et al., 2013; Arora et al., 2020; Boysen et al. 2020).

This analysis uses 15 CMIP6 models (Supplementary Table 1). Responses are estimated from years 101-140 ($CO_2$ quadruples in year 140) in the 1PCTCO2-bgc runs relative to the corresponding 40 years in the preindustrial control simulation. I refer to this 40-year time period as the time of $CO_2$ quadrupling. Preindustrial control simulations feature fixed (to the preindustrial value) atmospheric $CO_2$ concentration and other climate drivers (e.g., other greenhouse gases, solar irradiance, aerosols). Monthly mean data is used and all data is interpolated to a 2.5°x2.5° grid and aggregated to annual means. Only two models, GFDL-ESM4 and MPI-ESM1-2-LR, include dynamic vegetation (i.e., vegetation type and coverage can respond to the elevated $CO_2$). Three models, GFDL-ESM4, UKESM1-0-LL and NorESM2-LM, include atmospheric chemistry with an interactive representation vegetation biogenic volatile organic compound (BVOC) feedbacks (e.g., Gomez et al., 2023). Eight models, including





ACCESS-ESM1-5, CESM2, CMCC-ESM2, EC-Earth3-CC, MIROC-ES2L, MPI-ESM1-2-LR,
NorESM2-LM and UKESM1-0-LL, feature a terrestrial nitrogen cycle (Supplementary Table 1).
Additional model information can be found in Arora et al. (2020); Gomez et al. (2023); Allen et
al. (2024) and Gier et al. (2024).
**2.2 Surface Energy Balance Decomposition**
The Surface Energy Balance (SEB) decomposition (Luyssaert et al., 2014; Hirsch et al., 2018;
Boysen et al., 2020) is used to infer the contribution of changes in energy fluxes to changes in
surface temperature ($\Delta TS$):

$$\Delta TS = \frac{1}{4\epsilon\sigma TS^3_{control}}[\Delta SWD(1-\alpha) - \Delta\alpha(SWD) + \Delta LWD - \Delta LH - \Delta SH],$$


where ε is the surface emissivity assumed to be 0.97 (Boysen et al., 2020), σ is the Stefan-
Boltzmann constant with a value of $5.67 \times 10^{-8}$ W m$^{-2}$ K$^{-4}$, and TS$_{control}$ is the surface
temperature from the preindustrial control experiment. The first term in square brackets
represents the contribution from changes in downwelling surface shortwave radiation ($\Delta SWD$)
which is multiplied by the monthly mean climatology of (1-$\alpha$); the second term represents the
contribution from changes in surface albedo ($\alpha$) which is multiplied by the monthly mean SWD
climatology (changes in albedo impact upwelling surface shortwave radiation); the third term
represent the contribution from changes in downwelling surface longwave radiation ($\Delta LWD$);
the fourth term represents the contribution from changes in surface latent heat flux ($\Delta LH$); and
the final term represents the contribution from changes in surface sensible heat flux ($\Delta SH$). I
also decompose the first term on the right (i.e., the surface downwelling SW radiation term) into
the contribution from changes in surface downwelling shortwave radiation under clear-sky and
cloudy-sky conditions. The clear-sky contribution is estimated as $\Delta SWD_{clear}(1-\alpha)$, where
$\Delta SWD_{clear}$ is the change in clear-sky downwelling surface solar radiation. The cloudy-sky
contribution is estimated as the residual between the all-sky and clear-sky SWD radiation SEB
components. A similar decomposition is performed for downwelling surface longwave radiation
to isolate its clear-sky and cloudy-sky contributions. The SEB decomposition is performed over
all land areas. I note that the SEB decomposition does not account for all factors, including for
example the ground heat flux and changes in subsurface heat storage (both of which are assumed
to be zero here), or changes in surface emissivity.
**2.3 Statistical Significance**
Statistical significance of a response is estimated using two approaches. In the first approach
(e.g., Fig. 1), the multimodel mean time series for the experiment and the control is calculated
and their difference is computed. A 2-tailed pooled t-test is used to assess significance of this
difference at the 90% confidence level with $n_1+n_2-2$ degrees of freedom ($n_1$ is the number of
years in the experiment and $n_2$ is the number of years in the control, i.e., 40 years each) using the
pooled variance $\frac{(n_1-1)S_1^2+(n_2-1)S_2^2}{n_1+n_2-2}$, where $S_1$ and $S_2$ are the sample variances. Significance of the
multimodel mean response relative to each individual model response (e.g., Supplementary
Table 2) is estimated by comparing the average of the individual model responses relative to its
uncertainty, estimated as $\pm1.65 \times SE$ (i.e., the 90% confidence interval). SE is the standard



error estimated as $\frac{1.65 \times \sigma}{\sqrt{m}}$, where σ is the standard deviation across models and *m* is the number of
models. Model agreement on the sign of the multimodel mean response is spatially and globally
estimated as the percentage of models that yield a positive or negative response. A 2-tailed
binomial test yields model agreement at the 90% confidence level when at least 11 of the 15
(73%) models agree on the sign of the response. Significance of correlations (*r*) is estimated
from a two-tailed t-test as: $t = \frac{r}{\sqrt{\frac{1-r^2}{N-2}}}$, with *N-2* degrees of freedom. N is either the number of
grid boxes (for a spatial correlation) or the number of models (for correlations across models).

3. **Results**

**3.1 Vegetation, Land Carbon and Inferred Biogeochemical Temperature Responses**
Figure 1 shows multimodel mean annual mean responses of vegetation including NPP and LAI.
The global mean increase in NPP is $679.4 \pm 140.6$ kg km$^{-2}$ day$^{-1}$ (77.4% increase relative to the
control) with larger increases in the tropics (30S-30N) at $1049.7 \pm 248.2$ kg km$^{-2}$ day$^{-1}$ as
compared to the extratropics (30S-60S and 30N-60N) at $516.7 \pm 109.4$ kg km$^{-2}$ day$^{-1}$
(Supplementary Table 2). In each of these regions, all 15 models agree on a positive NPP
response (Supplementary Figure 1 shows the spatial model agreement on the sign of the
response). Similar results exist for LAI with a multimodel mean global mean increase of $0.71 \pm$
$0.25$ (Fig. 1b; 48.9% increase relative to the control), which increases to $1.12 \pm 0.41$ in the
tropics. Here, however, one model features a decrease in LAI (GISS-E2-1-G). One of the two
models with interactive vegetation, GFDL-ESM4, yields relatively large global mean NPP
increases (2$^{nd}$ largest) at $1207.3 \pm 30.9$ kg km$^{-2}$ day$^{-1}$ (global mean) whereas MPI-ESM1-2-LR
yields $673.2 \pm 13.9$ kg km$^{-2}$ day$^{-1}$, a value close to the multimodel global mean increase. In
terms of LAI, GFDL-ESM4 yields a value close to the multimodel global mean at $0.84 \pm 0.02$
while MPI-ESM1-2-LR yields the weakest global mean increase at $0.17 \pm 0.01$. Thus, there are
not clear model differences in these vegetation responses between those models with interactive
vegetation versus those models that lack interactive vegetation.

In terms of land carbon (cLand), the multimodel annual mean global mean increase is $4.52 \pm$
$0.68$ kgC km$^{-2}$ and all 14 models (GISS-E2-1-G is missing) agree on enhanced land carbon
sequestration. Decomposing land carbon into vegetation, soil organic matter and litter carbon
shows that the bulk of this increase is due to an increase in vegetation carbon (Fig. 1e) at $2.48 \pm$
$0.42$ kgC km$^{-2}$ (75.1% increase relative to the control). This is followed by an increase in soil
organic matter carbon (Fig. 1d) at $1.38 \pm 0.49$ kgC km$^{-2}$ (15.0% increase relative to the control)
and litter carbon (Fig. 1c) at $0.66 \pm 0.22$ kgC km$^{-2}$ (64.4% increase relative to the control).
Converting the above land carbon responses into multimodel mean global totals yields $468.1 \pm$
$89.4$ PgC; $248.0 \pm 89.5$ PgC; and $119.1 \pm 45.2$ PgC for vegetation carbon, soil carbon and
litter carbon, respectively. Thus, the total land carbon increase is $835.3 \pm 134.3$ PgC. Increases
in vegetation carbon contribute 56% to this value, followed by soil carbon at 30% and litter
carbon at 14%. Once again, models with interactive vegetation do not stand out, as GFDL-
ESM4 yields a total land carbon increase of $958.6 \pm 32.6$ PgC and MPI-ESM1-2-LR yields
$530.7 \pm 12.0$ PgC (4$^{th}$ smallest increase).



I separate the models into the eight (N models) that have a representation of the terrestrial
nitrogen cycle versus the six models (noN models) that do not (GISS-E2-1-G is missing so there
are only 14 models). As noted in Gier et al. (2024), the inclusion of nitrogen limitation led to a
large improvement in photosynthesis compared to models not including this process.
I find significantly larger increases in land carbon storage in noN models at $1153.4 \pm 213.6$ PgC
versus $581.6 \pm 157.5$ PgC in N models (consistent with Arora et al., 2020). Similar but less
significant statements apply for NPP ($923.4 \pm 202.2$ kg km$^{-2}$ day$^{-1}$ in noN models versus
$679.4 \pm 140.6$ kg km$^{-2}$ day$^{-1}$ in N models) and LAI ($0.94 \pm 0.48$ in noN models versus $0.62 \pm$
$0.24$ in N models). The weaker increase in land carbon storage in models with a terrestrial
nitrogen cycle is consistent with terrestrial nitrogen generally reducing the response of NPP and
carbon storage to elevated levels of atmospheric $CO_2$ because of an increasing limit of nitrogen
availability for carboxylation enzymes and new tissue construction (e.g., Jones et al., 2016).
I use the TCRE (Gillett et al., 2013; Arora et al., 2020; Boysen et al. 2020) to estimate the near-
surface air temperature (TAS) response to the aforementioned changes in land carbon
(biogeochemical effects). The TCRE quantifies the amount of warming relative to the
preindustrial state per unit cumulative emissions at the time when atmospheric $CO_2$
concentration doubles in the 1PCTCO2 simulation. The best estimate of the TCRE at 1.65 K per
1000 PgC, with a likely range from 1.0 to 2.3 K per 1000 PgC (Canadell et al., 2021) yields a
biogeochemical cooling effect of $-1.38$ ($-1.92$ to $-0.84$) K. Models without a terrestrial
nitrogen cycle (consistent with their enhanced land carbon storage) yield larger cooling (but not
significantly so) at $-1.90$ ($-2.65$ to $-1.15$) K relative to the models with a terrestrial nitrogen
cycle at $-0.96$ ($-1.34$ to $-0.58$) K. Similar biogeochemical cooling is obtained if I use an
estimate of each model's TCRE (as opposed to the best estimate) at $-1.22$ K for all models;
$-1.70$ K for noN models; and $-0.86$ K for N models. Thus, the cooling difference between
models with and without a terrestrial nitrogen cycle is largely due to differences in the land
carbon response.
Thus, carbon fertilization at the time of $CO_2$ quadrupling yields large increases in land carbon
storage and corresponding global mean biogeochemical cooling as inferred from the TCRE.
Models that lack a terrestrial nitrogen cycle tend to yield larger increases in land carbon storage
and in turn, larger inferred cooling, implying they may overestimate the magnitude of this
cooling effect. I note that the magnitude of this multimodel biogeochemical mean cooling is
relatively large, e.g., it is about 35% of the global mean warming under 1PCTCO2 and
1PCTCO2-rad (which only accounts for the radiative effect of $CO_2$) of $3.94 \pm 0.37$ K and
$3.78 \pm 0.35$ K, respectively.
**3.2 Biogeophysical Temperature Responses**
Figure 2a shows the multimodel annual mean near-surface air temperature response. As noted
above, the simulated temperature responses in these simulations includes only the biogeophysical
(non-carbon cycle) effects. The multimodel annual mean global mean TAS response is $0.16 \pm$
$0.09$ K with 13 of the 15 models yielding warming (Fig. 2b shows the spatial model agreement
on the sign of the response). The largest warming occurs in EC-Earth3-CC at $0.51 \pm 0.05$ K,
followed by CESM2 at $0.44 \pm 0.04$ K and UKESM1-0-LL at $0.40 \pm 0.04$ K (Supplementary
Figure 2). The two models that yield cooling are CMCC-ESM2 and CNRM-ESM2-1 at
$-0.01 \pm 0.07$ (not significant at the 90% confidence level) and $-0.31 \pm 0.04$ K, respectively





(Supplementary Figure 2e,f). Over land, the multimodel mean warming increases to $0.28 \pm 0.13$

K (14 of the 15 models yield warming). In both cases, warming is larger in the extratropics as
compared to the tropics (Supplementary Table 3). Thus, biogeophysical effects of carbon
fertilization yield warming, but much less as compared to the corresponding biogeochemical
effects noted above at $-1.38$ ($-1.92$ to $-0.84$) K. I also note that the biogeophysical warming
of carbon fertilization is much smaller as compared to the biogeophysical warming associated
with the radiative effects (from 1PCTCO$_2$-rad simulations) of CO$_2$ at $3.78 \pm 0.35$ K.

**3.3 Drivers of the Biogeophysical Temperature Response**
I use the surface energy balance (SEB; Section 2.2) decomposition (Luyssaert et al., 2014;
Hirsch et al., 2018; Boysen et al., 2020) to understand the drivers of the biogeophysical
temperature changes. I first note that the SEB decomposition reasonably reproduces the change
in surface temperature (TS) and TAS. For example, the SEB reconstructed multimodel annual
mean global land mean TS response is $0.33 \pm 0.13$ K relative to the actual TS and TAS
responses of $0.26 \pm 0.12$ K and $0.28 \pm 0.13$ K, respectively (Supplementary Table 4).

Figure 3 shows multimodel annual mean spatial responses for the main terms of the SEB
decomposition (Supplementary Figure 3 shows the corresponding model agreement on the sign
of the responses). The surface latent heat flux (LH) term (Fig. 3d) at $0.27 \pm 0.11$ K
(Supplementary Table 4) contributes the most to the global land warming, followed by the
downwelling surface longwave radiation (LW) term (Fig. 3c) at $0.20 \pm 0.13$ K. The surface
albedo ($\alpha$) term (Fig. 3a) contributes $0.11 \pm 0.06$ K and the downwelling surface shortwave
radiation (SW) term (Fig. 3b) contributes $0.09 \pm 0.06$ K. In contrast, the surface sensible heat
flux (SH) term (Fig. 3e) leads to cooling at $-0.34 \pm 0.08$ K. Model agreement on the sign of the
multimodel mean response occurs in 12 to 13 of the 15 models (depending on the SEB term;
Supplementary Table 4). As with the global land mean, the LH SEB term at $0.45 \pm 0.15$ K
contributes the most to the tropical land mean warming (14/15 models agree on warming). Over
extratropical land, the dominant SEB terms are the LW term ($0.20 \pm 0.16$ K), as well as the
surface albedo ($0.16 \pm 0.09$ K) and SW terms ($0.16 \pm 0.10$ K).

The large warming due to the LH SEB term, which can be decomposed into canopy transpiration
and evaporation (which includes sublimation) SEB terms, is consistent with decreased canopy
transpiration due to decreased stomatal conductance under the higher atmospheric CO$_2$
concentration, i.e., more efficient stomata that lose less water to the atmosphere (e.g., Wong et
al., 1979; Keenan et al., 2013). Figure 4 shows additional terms from the SEB decomposition,
including relatively large values for the transpiration SEB term (Fig. 4e; corresponding model
agreement spatial maps are included in Supplementary Figure 4). The multimodel global land
mean increase in the canopy transpiration SEB term is $0.45 \pm 0.15$ K (13 of the 13 models agree
on the increase; Supplementary Table 4), which increases to $0.70 \pm 0.26$ K in the tropics.
Warming associated with decreased canopy transpiration is muted to some extent through
increases in evaporation (which cools). The corresponding evaporation SEB term over global
land yields cooling of $-0.19 \pm 0.16$ K, but with reduced model agreement on the cooling (9 of
13 models). This again increases in magnitude over tropical land to $-0.26 \pm 0.25$ K but with
only 7 of 13 models agreeing on the cooling. The enhanced evaporation appears to be directly
related to the decrease in transpiration. Spatially correlating the multimodel mean change in the
transpiration and evaporation SEB terms yields a very strong global land correlation of $-0.83$,





which increases to −0.86 over tropical land. Similar results exist across models, i.e., the
correlation between each model's transpiration and evaporation SEB terms is −0.69 over global
land, which increases to −0.78 over tropical land (correlations are significant at the 99%
confidence level). This suggests that for conditions of reduced transpiration, evaporation
increases to try to satisfy the evaporative demand of the atmosphere.
Although all of the SEB terms may contain temperature induced feedbacks to some extent (since
these are coupled ocean-atmosphere simulations), warming associated with the LW SEB term
(particularly its clear-sky component) is likely a response to the surface warming as opposed to a
driver of the warming. Surface warming will lead to an increase in upwelling LW radiation
consistent with the Stefan-Boltzmann law whereby a blackbody radiates energy proportional to
$TS^4$. Some of this enhanced upwards longwave radiation (via the atmospheric greenhouse effect)
will be reradiated back down to the surface, i.e., enhanced downwelling LW radiation at the
surface. This argument is consistent with Vargas Zeppetello et al. (2019), who found surface
downwelling LW radiation is tightly coupled to surface temperature. Changes in the LW SEB
term are also likely augmented by increases in atmospheric water vapor via the water vapor
feedback (i.e., a stronger greenhouse effect). For example, the multimodel global mean
tropospheric specific humidity significantly increases by $0.017 \pm 0.014$ g kg$^{-1}$ (increases also
occur over land but these are not significant; Supplementary Table 3; Supplementary Figure 5f).
I also note that the cooling under the SH SEB term is likely in part a feedback to the surface
warming as well as a response to the warming under the LH SEB term. In the former case,
surface warming (largely induced by decreases in surface latent heat flux, decreases in surface
albedo and increases in solar radiation) will lead to an increase in sensible heat flux, which will
act to cool the surface. In the latter case, a reduction in latent heat flux will be compensated by
an increase in sensible heat flux. Spatially correlating the multimodel mean change in the SH
and LH SEB terms yields a very strong global land correlation of −0.90, which increases to
−0.92 over tropical land. Similar results exist across models, i.e., the correlation between each
model's SH and LH SEB terms is −0.84 over global land, which increases to −0.86 over
tropical land (correlations are significant at the 99% confidence level). Thus, these correlations
support an inverse relationship between the SH and LH SEB responses. As the LH SEB term
leads to warming consistent with reduced stomatal conductance under elevated $CO_2$, this is in
part compensated for by SH SEB cooling.
The LW SEB term can be decomposed into clear-sky (LW$_{clear}$; Fig. 4c) and cloudy-sky (LW$_{cloud}$;
Fig. 4d) contributions. All of the multimodel mean global land warming associated with the LW
SEB term is due to the LW$_{clear}$ SEB component at $0.27 \pm 0.13$ K (12 of 14 models agree on the
warming). The dominance of the LW$_{clear}$ SEB term warming is consistent with the
aforementioned greenhouse effect, combined with increased atmospheric water vapor. In
contrast, the LW$_{cloud}$ SEB term contributes to multimodel mean global land cooling of $-0.07 \pm$
0.03 K (12 of 14 models agree on the cooling). This cooling is consistent with a multimodel
mean global land decrease in total cloud cover at $-0.52 \pm 0.23$ % (12 of 15 models agree on the
decrease; Supplementary Figure 5a and 6a). A decrease in cloud cover will act similarly to the
greenhouse effect (i.e., here a weaker greenhouse effect) and this will promote a decrease in
surface downwelling LW radiation. Consistently, larger multimodel mean LW$_{cloud}$ SEB cooling



occurs in the extratropics at $-0.11 \pm 0.05$ K, consist with the larger decrease in extratropical
total cloud cover at $-0.64 \pm 0.29$ %.
I note that the multimodel mean decrease in global land cloud cover is consistent with the
previously discussed decrease in latent heat flux (largely due to decreases in canopy
transpiration) and with decreases in near-surface and tropospheric relative humidity over land
(Supplementary Figure 5c,d and 6c,d). For example, the multimodel mean global land decrease
in near-surface relative humidity is $-0.87 \pm 0.48$ %, which increases in magnitude (as does
canopy transpiration) to $-1.09 \pm 0.59$ % over tropical land (11 of 14 models agree on both
decreases). The corresponding decrease in multimodel mean global land tropospheric relative
humidity is weaker but still significant at $-0.21 \pm 0.12$ % (12 of 14 models agree on the
decrease). The near-surface relative humidity decrease over land is consistent with near-surface
land warming (which increases the water vapor carrying capacity of the air) and with a (non-
significant) decrease in near-surface specific humidity ($-0.010 \pm 0.052$ g kg$^{-1}$; Supplementary
Figure 5e and 6e). The tropospheric relative humidity decrease over land is consistent with
tropospheric warming ($0.18 \pm 0.09$ K) dominating over non-significant increases in tropospheric
specific humidity ($0.011 \pm 0.015$ g kg$^{-1}$; Supplementary Figure 5f and 6f). In other words, the
thermodynamic increase in water vapor over land as dictated by the Clausius Clapeyron equation
(e.g., water vapor increases by 7% per K of warming) is muted by the decrease in latent heat
flux, and in particular canopy transpiration. I also note a multimodel mean decrease in global
land precipitation (Supplementary Figure 5b and 6b) at $-0.028 \pm 0.017$ mm day$^{-1}$ ($-1.2$ %
change relative to the control) with 12 of 15 models agreeing on the decrease.
Warming associated with the surface albedo SEB term is consistent with surface darkening (e.g.,
Betts et al., 2000, Bala et al., 2006; Li et al., 2015) under enhanced vegetation (e.g., LAI
increases; Fig. 1b). In contrast to maximum tropical warming due to the LH SEB term, warming
associated with the surface albedo SEB term is largest in the extratropics. The corresponding
multimodel mean extratropical land warming is $0.16 \pm 0.09$ K relative to the tropical warming
of $0.06 \pm 0.04$ K. Larger extratropical as opposed to tropical warming under the surface albedo
SEB term is consistent with larger vegetation-induced darkening over higher latitudes, where
snow and ice (bright surfaces with high surface albedo) are more prevalent.
The SW SEB term, which yields multimodel mean global land warming of $0.09 \pm 0.06$ K (12 of
15 models agree on the warming), can be decomposed into clear-sky (SW$_{clear}$; Fig. 4a) and
cloudy-sky (SW$_{cloud}$; Fig. 4b) contributions. In contrast to the total SW SEB term, the SW$_{clear}$
SEB term yields multimodel mean global land cooling at $-0.05 \pm 0.02$ K. Such cooling is
consistent with the aforementioned increase in tropospheric specific humidity, which increases
the atmospheric absorption of shortwave radiation by water vapor. Changes in atmospheric
aerosols may also contribute though direct scattering/absorption of solar radiation. Few models,
however, archived the relevant aerosol diagnostics and changes in the multimodel mean aerosol
optical depth (AOD; Supplementary Figure 7) lack significance except for tropical land
(Supplementary Table 3). For example, a nonsignificant multimodel global land AOD increase
of $1.88 \pm 2.50$ $10^{-3}$ occurs, with 5 of the 8 models agreeing on the increase. This increases and
becomes significant over tropical land at $3.07 \pm 3.00$ $10^{-3}$ (5 of 8 models agree on the increase).
Part of these AOD increases are due to (nonsignificant) increases in dust AOD (DAOD;
Supplementary Table 3). If I remove DAOD from (total) AOD (i.e., $AODNOD = AOD -$





$DAOD$), I find nonsignificant AODNOD increases for global and extratropical land, but
significant increases over tropical land at $2.35 \pm 2.33 \ 10^{-3}$ (5 of 8 models agree on this increase).
Of the three models with an interactive representation of BVOCs, only two archived AOD
(GFDL-ESM4 and UKESM1-0-LL), and both models yield much larger AOD increases than the
other models. Such AOD increases are consistent with enhanced BVOC emissions due to the
increased vegetation (e.g., LAI; Fig. 1b), leading to more secondary organic aerosol (e.g., Scott
et al., 2018; Weber et al., 2024). For example, AOD increases over global land by $7.78 \pm 1.00$
$10^{-3}$ and $7.87 \pm 0.78 \ 10^{-3}$ for GFDL-ESM4 and UKESM1-0-LL, respectively (compared to the
multimodel mean increase of $1.88 \pm 2.50 \ 10^{-3}$). For GFDL-ESM4, a large fraction of this AOD
increase is due to an increase in DAOD at $4.82 \pm 0.91 \ 10^{-3}$ (UKESM1-0-LL yields a
nonsignificant DAOD decrease of $-0.38 \pm 0.78 \ 10^{-3}$). Nonetheless, AODNOD (which includes
SOA) yields relatively large and significant global land increases for both models at $2.97 \pm 0.41$
$10^{-3}$ for GFDL-ESM4 and $8.26 \pm 0.38 \ 10^{-3}$ for UKESM1-0-LL (Supplementary Figure 7g-h).
These values increase over tropical land (where vegetation indices also increase the most) at
$6.16 \pm 0.85 \ 10^{-3}$ and $10.68 \pm 0.63 \ 10^{-3}$ for GFDL-ESM4 and UKESM1-0-LL, respectively. In
turn, both models feature $SW_{clear}$ SEB cooling (consistent with enhanced aerosol scattering) over
global land (and over tropical and extratropical land) of $-0.13 \pm 0.02$ for GFDL-ESM4 and
$-0.07 \pm 0.02$ K for UKESM1-0-LL (compared to the multimodel mean of $-0.05 \pm 0.02$ K).
NorESM2-LM, the other model with an interactive representation of BVOCs (but no AOD data),
also yields relatively large $SW_{clear}$ SEB cooling at $-0.08 \pm 0.03$ K. The GFDL-ESM4 $SW_{clear}$
cooling is the second largest of the 15 models; the NorESM2-LM and UKESM1-0-LL $SW_{clear}$
cooling are third and fourth largest, respectively. Similar statements also generally apply for
tropical land, e.g., the NorESM2-LM $SW_{clear}$ cooling of $-0.16 \pm 0.05$ K is the largest and the
GFDL-ESM4 $SW_{clear}$ cooling of $-0.11 \pm 0.03$ K is the third largest; however, UKESM1-0-LL
$SW_{clear}$ cooling is not exceptional at $-0.04 \pm 0.04$ K (compared to the multimodel mean of
$-0.05 \pm 0.02$ K). Thus, consistent with Gomez et al. (2023), there is evidence that models with
interactive chemistry yield AOD increases under carbon fertilization, consistent with enhanced
vegetation leading to more BVOC emissions and SOA. In turn, this appears to strengthen the
cooling associated with the $SW_{clear}$ SEB term (with enhanced water vapor and reduced surface
solar radiation also contributing). I also note that the AOD increase here is also consistent with
the land-sea warming contrast (e.g., Fig. 2a) and reduced precipitation over land (Supplementary
Figure 5b), which leads to less aerosol wet removal (Allen et al., 2019).
As the $SW_{clear}$ SEB term leads to multimodel mean cooling, the warming under the (total) SW
SEB term is therefore associated with clouds. $SW_{cloud}$ yields multimodel mean global land
warming of $0.14 \pm 0.05$ K (12 of 15 models agree on the increase), which increases to $0.21 \pm$
$0.08$ K over extratropical land. As with the $LW_{cloud}$ SEB cooling, this $SW_{cloud}$ SEB warming is
consistent with decreases in cloud cover (e.g., here, less cloud cover will lead to enhanced
surface solar radiation and warming). Moreover, both the $LW_{cloud}$ and $SW_{cloud}$ SEB terms are
largest in magnitude in the extratropics, consistent with larger extratropical cloud cover decrease.
I note that the net effect of clouds is controlled by their impact on shortwave as opposed to
longwave radiation. The $SW_{cloud}$ SEB term yields multimodel global land mean warming of
$0.14 \pm 0.05$ K, while the $LW_{cloud}$ SEB term yields corresponding cooling of $-0.07 \pm 0.03$ K.
This implies the net effect of clouds is associated with warming of $0.07 \pm 0.06$ K. The net
warming effect from clouds increases over extratropical land at $0.10 \pm 0.09$ K.




I also note that I do not find evidence for an aerosol indirect effect on clouds and in turn the
SW$_{cloud}$ SEB term. As discussed above, the two models with interactive chemistry simulate
relatively large and significant increases in AODNOD, particularly over tropical land (which
potentially leads to SW$_{cloud}$ cooling related to cloud brightening and enhanced cloud lifetime).
However, as with the multimodel mean, SW$_{cloud}$ leads to warming over global land, tropical land
and extratropical land for both models (as well as NorESM2-LM). UKESM1-0-LL actually
yields the largest SW$_{cloud}$ warming of the 15 models over both global land and tropical land at
$0.32 \pm 0.02$ K and $0.28 \pm 0.05$ K, respectively. In contrast to an aerosol indirect effect, this
SW$_{cloud}$ warming in UKESM1-0-LL is consistent with a decrease in cloud cover, as UKESM1-0-
LL yields the second largest cloud cover decrease over global land at $-1.35 \pm 0.14\%$ and the
third largest decrease over tropical land at $-1.20 \pm 0.29\%$. Overall, the (total) SW SEB term
warms global land in UKESM1-0-LL and NorESM2-LM by $0.25 \pm 0.03$ K and $0.10 \pm 0.05$ K,
respectively, with nonsignificant cooling of $-0.02 \pm 0.02$ K for GFDL-ESM4. Thus, as with
the multimodel mean results, SW$_{cloud}$ SEB warming dominates over SW$_{clear}$ SEB cooling in these
three models. Any impacts of interactive chemistry and enhanced BVOC emissions/SOA on
clouds appear to be minor in these simulations and do not lead to an appreciable increase in
cooling.

Spatially correlating the multimodel mean SEB responses with the corresponding total SEB
response yields significant global (land only) correlations for all individual SEB terms
(Supplementary Table 5). Consistent with the sign of the multimodel mean SEB responses, the
total SEB term is positively correlated with the surface albedo, SW, SW$_{cloud}$, LW, LW$_{clear}$, LH
and canopy transpiration SEB terms. Similarly, the total SEB term is negatively correlated with
the SW$_{clear}$, LW$_{cloud}$, SH and evaporation SEB terms. The LW$_{clear}$ SEB term yields the largest
global correlation with the total SEB term at 0.91. This very high correlation adds additional
evidence that the LW$_{clear}$ SEB term is largely a feedback to the surface warming. Restricting this
analysis to the tropics yields maximum correlations between the total SEB term and the LH SEB
term at 0.77, followed closely by the LW$_{clear}$ SEB term at 0.76. This provides additional support
for the importance of the multimodel mean LH SEB response to the total SEB response in the
tropics. In the extratropics, maximum correlations occur for LW$_{clear}$ at 0.90, followed by LW at
0.58 and SW$_{cloud}$ at 0.57.

I conduct additional analyses to better understand causes of inter-model spread in the SEB
responses. Supplementary Figures 8 and 9 shows spatial correlations (across models) between
the total SEB term and each of the SEB components. The LW SEB term show very large and
significant correlations ($r > 0.80$) throughout most of the Northern Hemisphere (with nearly all
land areas exhibiting similarly large and significant correlations under LW$_{clear}$). The LH SEB
term shows relatively large and significant positive correlations ($r > 0.70$) throughout the tropics
(minus the Sahara Desert). The surface albedo SEB term shows relatively large and significant
positive correlations ($r > 0.70$) in the NH extratropics. This suggests much of the intermodel
variation in the total SEB extratropical surface temperature response is related to intermodel
differences in the SEB surface albedo term response; in the tropics, intermodel variation in the
total SEB tropical surface temperature response is related to intermodel differences in the SEB
surface LH term response.



These broad conclusions are consistent with Supplementary Figures 10-15, which show
corresponding model scatterplots between the total SEB term and its components for the global
land mean (Supplementary Figures 10-11), tropical land mean (Supplementary Figures 12-13)
and extratropical land mean (Supplementary Figures 14-15). The sign of all of the intermodel
correlations is consistent with the multimodel mean responses, i.e., positive intermodel
correlations (although many are not significant) between the total SEB response and the SEB
albedo, SW, LW, and LH responses occur. Similarly, a negative correlation between the total
SEB response and the SEB SH response occurs. A similar statement can be made for the
additional SEB terms where positive correlations exist between the total SEB term and the
$SW_{cloud}$, $LW_{clear}$, and canopy transpiration SEB terms, with negative correlations for the $SW_{clear}$
(except for the tropics), $LW_{cloud}$ and evpoaration SEB terms. Globally (Supplementary Figures
10-11), the intermodel variation in the total SEB response is largely related to the intermodel
variation in the SEB albedo response, with a correlation of 0.73 (significant at the 99%
confidence level). As noted above, CNRM-ESM2-1 is the lone model that yields significant
surface cooling (e.g., Supp. Fig. 2f) and this is the lone model that yields cooling for the SEB
albedo term. Furthermore, as noted above UKESM1-0-LL and EC-Earth3-CC yield relative
large total SEB warming, and they both possess relatively large SEB surface albedo warming.
The LW SEB term also yields a similar correlation at 0.73, which improves to 0.86 for $LW_{clear}$.
As noted above, however, I suggest this is largely a response to the warming and not a primary
driver. Interestingly, the correlation between the LH SEB term and the total SEB term is not
significant at 0.13. The lone model that yields cooling for the LH SEB term is EC-Earth3-CC,
and this is a model with relatively large total SEB warming. This implies intermodel variation in
the relative importance of the individual SEB responses their total SEB response.
In the tropics, the intermodel variation in the total SEB term is largely related to the SW SEB
term with a correlation of 0.70 (99%), which is largely due to $SW_{cloud}$ (r=0.74). The LH SEB
term is also important with a correlation at 0.54 (95%). In the extratropics, the intermodel
variation in the total SEB term is largely related to the SEB albedo term with a correlation of
0.81 (99%). The LW SEB term also yields a similar correlation at 0.79, which improves to 0.91
for $LW_{clear}$.
Finally, I note a significant correlation of 0.55 (significant at the 95% confidence level; Figure 5)
between the transient climate response (TCR; warming centered on the time of $CO_2$ doubling)
and the global mean biogeophyscial temperature response associated with carbon fertilization
across models. The correlation improves to 0.68 (significant at the 99% confidence level) over
global land only (Fig. 5b). In other words, models with a larger (smaller) TCR tend to have
larger (smaller) biogeophysical warming under carbon fertilization. This is not necessarily
unexpected, since the TCR is based on the 1PCTCO2 experiments, which include the 1PCTCO2-
bgc responses. Moreover, the causes of larger TCR (e.g., climate feedbacks including water
vapor, tropospheric lapse rate, surface albedo and clouds) also operate in the 1PCTCO2-bgc runs.
In particular, the three models (EC-Earth3-CC, UKESM1-0-LL and CESM2) mentioned above
that yield the largest biogeophysical warming associated with carbon fertilization are among the
models with the largest TCR (CanESM5 is an exception at it has a large TCR but relatively small
biogeophysical warming). On one hand, this implies the intermodel spread in the biogeophysical
warming associated with carbon fertilization is related to the TCR; on the other hand, it also
implies the importance of the biogeophysical warming associated with the carbon fertilization



effect to intermodel variation in the TCR. I reiterate, however, that the magnitude of
biogeophysical warming associated with carbon fertilization is relatively small. If I re-estimate
the global mean near surface air temperature response over the time of $CO_2$ doubling (years 60-
79, as with TCR), I find biogeophysical warming of $0.12 \pm 0.08$ K, i.e., about 6% as large as the
multimodel mean TCR of $1.97 \pm 0.20$ K. However, some models yield much larger
biogeophysical warming at the time of $CO_2$ doubling, including many of the same models
previously discussed. EC-Earth3-CC yields warming of $0.26 \pm 0.05$ K, which is 10% of its
TCR of $2.7 \pm 0.10$ K. CESM2 yields warming of $0.49 \pm 0.07$ K, which is 20% of its TCR of
$2.4 \pm 0.07$ K. UKESM1-0-LL yields a percentage closer to the multimodel mean, with warming
of $0.19 \pm 0.07$ K, which is 7% of its TCR of $2.7 \pm 0.14$ K. Additional discussion on potential
causes of intermodel differences is included in the Supplement (Supplementary Note 1).
**4. Conclusions**
Using 15 CMIP6 models, I show that carbon fertilization at the time of $CO_2$ quadrupling (and in
the absence of radiative warming from $CO_2$) yields biogeophysical global mean warming of
$0.16 \pm 0.09$ K with 13 of the 15 models yielding warming. This warming increases over global
land to $0.28 \pm 0.13$ K with 14 of the 15 models yielding warming (Supplementary Table 4).
Using the surface energy balance decomposition to understand the drivers of this warming shows
that it is largely related to decreased latent heat flux, which leads to global land warming of
$0.27 \pm 0.11$ K (13 of 15 models agree on the warming). This in turn is largely associated with
reduced canopy transpiration which leads to global land warming of $0.45 \pm 0.15$ (13 of 13
models agree on the warming). Such a response is consistent with reduced stomatal conductance
under elevated $CO_2$ (e.g., Wong et al., 1979; Keenan et al., 2013). To some extent, this warming
if offset by increases in evaporation, which leads to global land cooling of $-0.19 \pm 0.16$ K. This
cooling, however, is less robustly simulated with only 9 of the 13 models agreeing on the
cooling. In the tropics, the importance of transpiration induced warming increases to $0.70 \pm$
$0.26$ K (13/13 models agree). Tropical land evaporation increases only marginally to $-0.26 \pm$
$0.25$ K but with limited model agreement (7/13 models agree on the cooling). Of the various
SEB terms, the evaporation term is the most uncertain across the models (particularly in the
tropics) with limited model agreement on the sign of the response. Given the importance of LH
to the biogeophysical warming response (both its direct impact as well as its indirect impact on
clouds and subsequently $LW_{cloud}$ and $SW_{cloud}$) and the competing effects of transpiration versus
evaporation, the low model agreement on the sign of the evaporation response highlights an
important source of model uncertainty.
Other important drivers of biogeophysical warming under carbon fertilization, particularly in the
extratropics, include reduced albedo and enhanced SW radiation due to a decrease in cloud
cover. Warming due to reduced albedo is consistent with the surface darkening effect of
vegetation (e.g., Betts et al., 2000, Bala et al., 2006; Li et al., 2015), particularly at higher
latitudes where snow/ice is more prevalent. Warming due to enhanced shortwave radiation due
to clouds is consistent with decreases in total cloud cover. Similar to the decrease in latent heat
flux, the decrease in cloud cover is related to the reduced stomatal conductance, which is
associated with reduced relative humidity over land which promotes a decrease in cloud cover.
Models with interactive BVOCs yield a larger increase in AOD, which strengthens the cooling
associated with the $SW_{clear}$; however, as with the multimodel mean, $SW_{cloud}$ warming dominates



over SW$_{clear}$ cooling in these models, implying any aerosol effect is minor in these simulations
(i.e., the dominant SW effect is warming due to cloud cover reductions associated with decreases
in transpiration).
Intermodel variation in the vegetation and biogeophysical temperature responses was also
evaluated. Although there are significant differences between models with and without a
terrestrial nitrogen cycle (e.g., N models yields significantly less carbon storage), most of these
differences (e.g., N models tend to yield less biogeochemical cooling) are not significant.
However, intermodel biogeophysical warming significantly correlates with each model's TCR.
This implies the causes of intermodel TCR variation (i.e., climate feedbacks) are also responsible
for some of the intermodel spread in the biogeophysical temperature response under carbon
fertilization. Finally, the increase in land carbon storage under carbon fertilization was used to
estimate the biogeochemical cooling effect using the transient climate response to cumulative
emissions. I find that biogeochemical cooling of $-1.38$ K ($-1.92$ to $-0.84$ K) dominates over
biogeophysical warming, by about an order of magnitude.
**Code Availability**
Standard code (e.g., NCL) was used to analyze the model simulations.
**Data Availability**
CMIP6 data can be downloaded from the Earth System Grid Federation at
https://aims2.llnl.gov/search.
**Author Contributions**
RJA conceived the project, analyzed model simulations and wrote the manuscript.
**Competing Interests**
The author declares no competing interests.

**Acknowledgements**
R. J. Allen is supported by NSF grant AGS-2153486.
**Financial Support**
This research has been supported by the National Science Foundation (grant no. AGS-2153486).



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



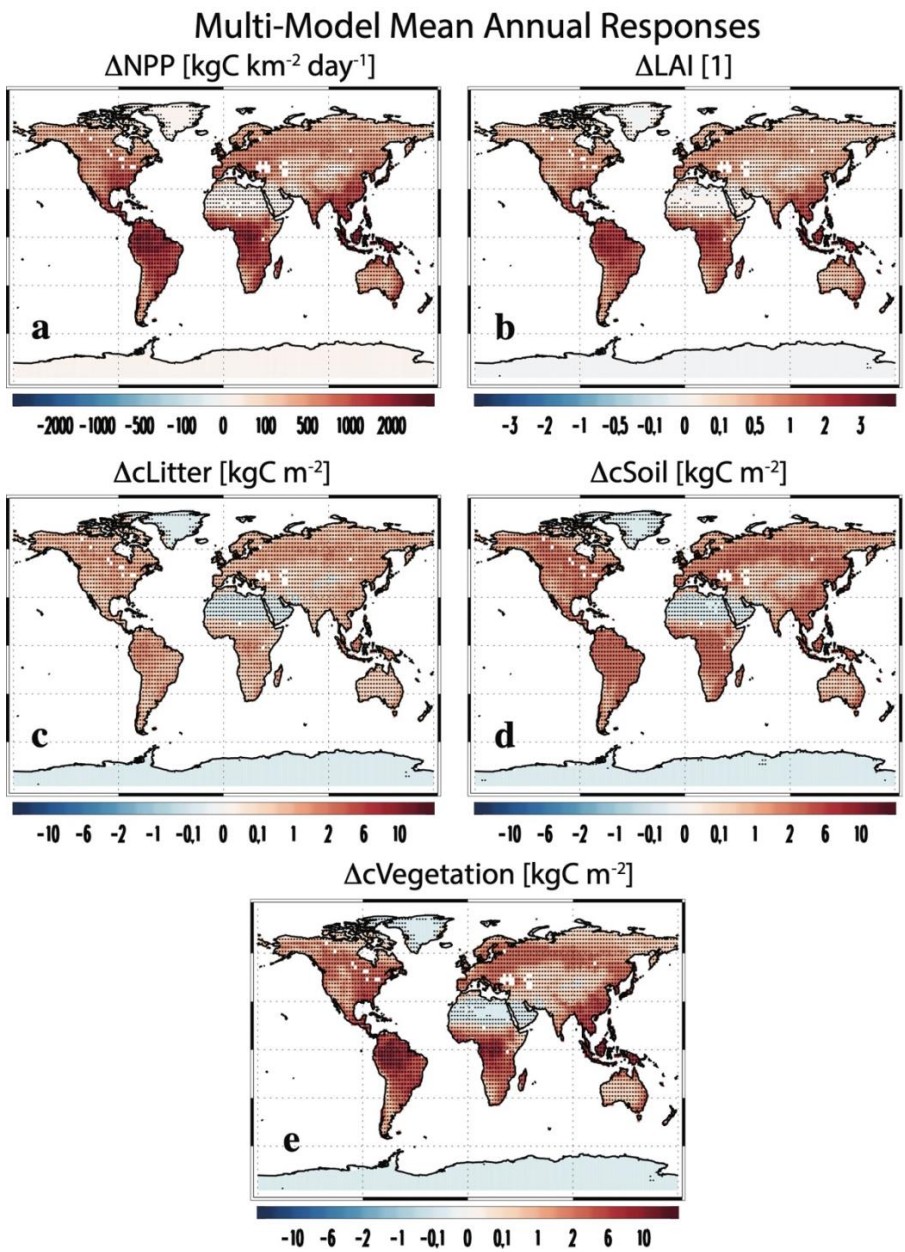

**Figure 1. Vegetation and land carbon responses.** Multimodel mean annual mean responses for (a) net primary productivity (NPP; kgC km$^{-2}$ day$^{-1}$); (b) leaf area index (LAI; dimensionless); (c) litter pool carbon (cLitter; kgC m$^{-2}$); (d) soil pool carbon (cSoil; kgC m$^{-2}$); and (e) vegetation carbon (cVegetation; kgC m$^{-2}$). Symbols denote a response significant at the 90% confidence level based on a two-tailed pooled t-test.



## Near Surface Air Temperature (TAS)

Multi-Model Mean Annual Response [K]    Model Agreement on Sign of Response [%]

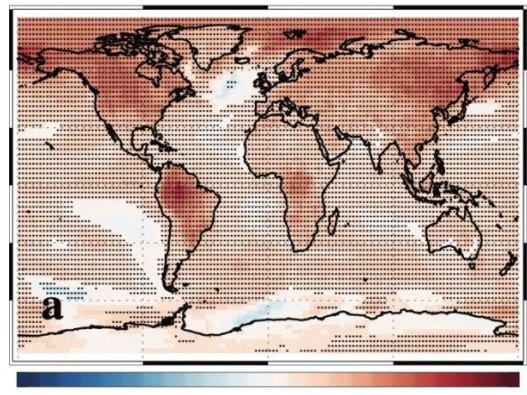
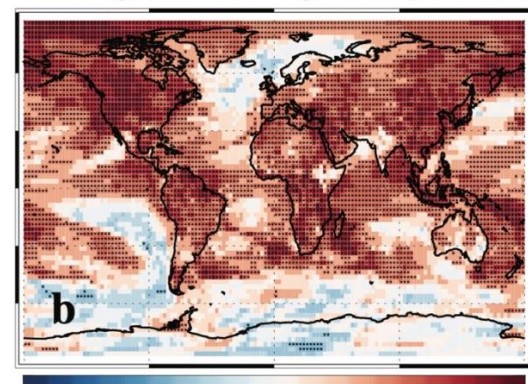

-0,9 -0,7 -0,5 -0,3 -0,1  0  0,1 0,3 0,5 0,7 0,9        -95-90-85-80-75-70-65-60-55  55 60 65 70 75 80 85 90 95

**Figure 2. Near surface air temperature.** (a) Multimodel mean annual mean near surface air
temperature (TAS; K) response. Symbols denote a response significant at the 90% confidence
level based on a two-tailed pooled t-test. (b) Model agreement on the sign of the TAS response
[% of models]. Red (blue) colors indicate model agreement on an increase (decrease). Symbols
represent significant model agreement at the 90% confidence level based on a two-tailed
binomial test.





**Figure 3. Surface energy balance (SEB) decomposition of the surface temperature**
**response**. Multimodel mean annual mean SEB responses for (a) surface albedo; (b) downwelling
surface shortwave radiation; (c) downwelling surface longwave radiation; (d) surface latent heat
flux; (e) surface sensible heat flux; and (f) the total (i.e., sum of the prior five terms). Units are
K. Symbols denote a response significant at the 90% confidence level based on a two-tailed
pooled t-test.



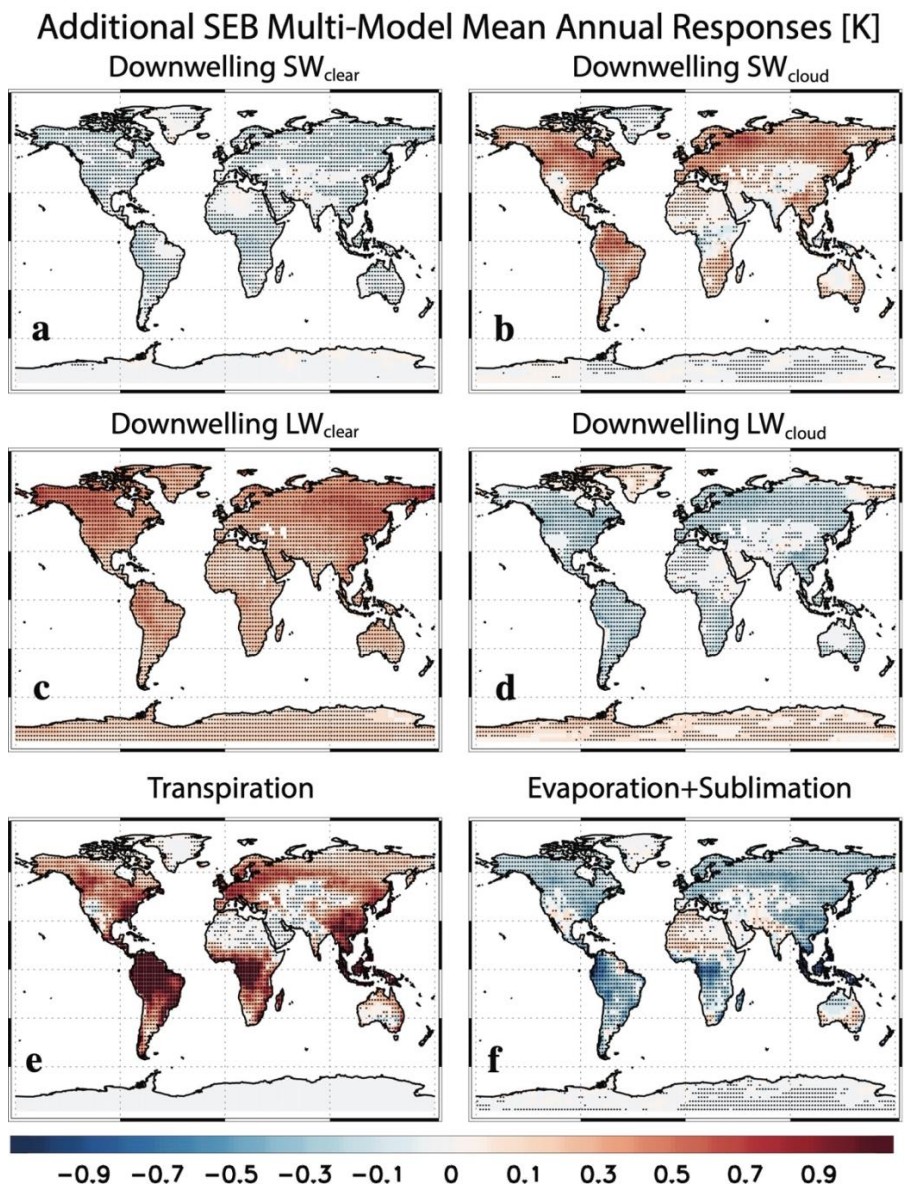

**Figure 4. Additional surface energy balance (SEB) decomposition of the surface**
**temperature response**. Multimodel mean annual mean SEB responses for downwelling surface
shortwave radiation decomposed into (a) clear-sky (SW$_{clear}$) and (b) cloudy-sky (SW$_{cloud}$)
contributions; downwelling surface longwave radiation decomposed into (c) clear-sky (LW$_{clear}$)
and (d) cloudy-sky (LW$_{cloud}$) contributions; and surface latent heat flux decomposed into (e)
canopy transpiration and (f) evaporation (which includes sublimation) contributions. Units are
K. Symbols denote a response significant at the 90% confidence level based on a two-tailed
pooled t-test.



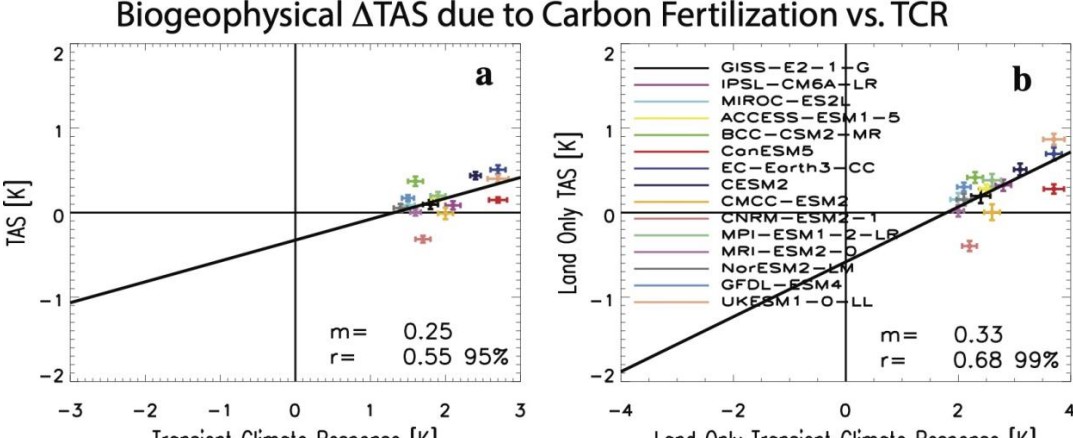

**Figure 5. Scatterplots of the global mean near-surface temperature response versus the transient climate response across models.** Scatterplots between the (a) global mean near-surface air temperature response (TAS, Y-axis) and the transient climate response (TCR). Panel b is analogous but for land only. Each symbol represents an individual model (see legend). Error bars for each symbol represent the 90% confidence intervals based on a two-tailed pooled t-test. Black line represents the least squares linear regression line. The corresponding slope (m) of the regression and the correlation coefficient (r) are included. Significant correlations based on a two-tailed test at the 95% and 99% confidence level are indicated. Units are K.