# Peer review of "The Biogeophysical Effects of Carbon Fertilization of the Terrestrial Biosphere"

_EGUsphere, 2025_

## Referee Comment (RC1)

**Review on**

**Robert J. Allen, "The Biogeophysical Effects of Carbon Fertilization of the Terrestrial Biosphere"**

**submitted to Atmospheric Chemistry and Physics (ACP).**

May 8, 2025

In simulations where the radiative effect of rising $CO_2$ is switched off, nevertheless a warming is observed. This warming is a consequence of the reaction of the biogeochemical reaction of the land biosphere on rising $CO_2$, namely the physiological effect of $CO_2$ fertilization. This goes along with a closure of stomata and thus reduced transpiration, as well as an increased plant productivity typically leading to an increased growth and thus a reduction in albedo. These effects – reduced transpiration and reduced albedo – together with their consequences (e.g. changed cloudiness) explain the warming observed in those simulations as is known at least since the pioneering study by Bala et al., 2006, soon followed by investigations with other models (e.g. [1, 2] and Cao et al., 2010). The new study by Allen re-investigates this behaviour taking advantage of the recent CMIP6 multi-model ensemble. I don't have a complete overview on the literature on the subject, but I suspect that the new aspects of this study are the comparative investigation of results from a rather large ensemble of models, and methodologically the usage of an energy balance decomposition as main tool.

Overall the study is well written, I like in particular the rather concise summary in the "Conclusions" section. Nevertheless, the mentioning of so many numbers in the text and the usage of so many abbreviations makes reading tedious. I personally would have preferred to leave numbers to tables or illustrative figures and restrict the text to qualitative discussions – I don't think that in this way the scientific content would deteriorate. Moreover, I am not really sure whether all the figures are really needed, in particular since from the many geographic details seen in the maps almost nothing is discussed. Maybe a visualization of the size of the different terms of the energy balance would be more illuminating than the many maps. But this is largely a matter of style to be decided by the author.

As far as I see, the results of this new study only confirm the insights obtained in previous investigations. Nevertheless, I think that because of the large range of models included in the analysis the paper could get an important reference on the subject. But for this a thorough review of the results of previous studies on the subject and also a comparison with their results should be added. Otherwise I fully support the publication of this study.

**Major comments**

**1.** The abstract should be more concise: Why do we need another study on the biogeochemical effects of carbon fertilization? What is specific about this study? I think one should also mention that an energy decomposition is used to disentangle the different effects on the surface energy budget.

**2.** The text in line 86ff leaves the impression that the author is the first to find biogeo-physically induced global warming. But this is not the first study quantifying the climate change caused by the physiological effects of $CO_2$ on plants (see above). Results of these and similar studies should be reported in the introduction. Moreover, the literature references on the subject should be completed (two additional references are found below, but I guess one finds more if one follows the citations).

**3.** Completely missing is a comparison of the results of this study with those obtained in earlier studies on the same topic.

**3.** Names for the energy balance terms are not explicitly introduced and moreover those names are not consistently used. E.g. the term from latent heat is called "LH SEB term" in the text, "Latent Heat" in the titles of Fig. 3 and Supplementary Fig. 8, and "LH" in Supplementary tables 4 and 5. Because most of the paper centers around the discussion of these terms, I suggest that names for the different terms are explicitly introduced early in the paper, maybe already when the energy balance equation is introduced in section 2.2.

**4.** Today all data sets have a unique identification by a DOI. Such DOIs should be provided for all data sets used.

**5.** The four paragraphs in lines 473-529 are discussing exclusively material from the Supplement. In order that the main text is self-contained this material should either be included in the main text or that whole text passage should be moved to the Supplement.

**Minor comments**

- First sentence of abstract is inappropriate here: Here it is stated that $CO_2$ fertilization is a "significant source of uncertainty in future climate projections". This is indeed correct, but the sentence leaves the impression as if the study could contribute to a reduction of this uncertainty. But this is not the case, so that the study cannot be motivated in this way.
- Line 49: Preposition missing: "...associated with ..." or "...associated to the ...". But even better would be to name also the direction of causality, e.g. "...increasing atmospheric $CO_2$ concentrations lead to carbon fertilization ...".
- Line 64: I find it a bit misleading to denote the cooling induced by enhanced carbon storage as a "biogeochemical" effect. I would say that the cooling is caused physically (its a radiative effect) even though its induced biogeochemically.
- Line 85: What does "Climate effects include ...as well as the drivers" mean? What "drivers" are "climate effects"?
- Lines 137-138: Something is wrong with the grammar.
- Line 138: The word "including" is superfluous since all eight models are explicitly named.
- Lines 206-208: What means "not clear model differences"? It seems that the author expects that the dynamic vegetation modules have a particular impact on the changes

in LAI, or what is the reason to consider here in particular the two models with dynamic vegetation?

- Line 304: The wording "more efficient stomata" is a bit weird, more efficient are not the stomata but is the water usage.
- Lines 317-320 (similarly lines 344 and 488): I am not sure what exactly the reported "results . . . across models" are. I can imagine that the author has analyzed the correlation between transpiration and vaporation for each model separately, but if so I had expected that the author reports a mean correlation value *plus* a range indicating the spread across the different models, but in the text one finds only single numbers for the global and tropical regions.
- Line 334: The remark that increases occur "also" over land is only understandable when consulting te respective figure in the Supplement: Indeed from that figure it is visible that significant changes are mostly found over oceanic regions, which explains the "also". But the main text should stand for its own without such a consultation.
- Lines 337-342: I cannot really follow this passage: What is called here the "former case" doesn't look for me like a "feedback" via the atmosphere but simply as a consequence of the changed temperature gradient due to the surface warming. And concerning the "latter case", the author seems to think that the mentioned "compensation" would be a process different from the mechanism described in the "former case", but I doubt that this can be justified by physics: "compensation" is not really a physical process but a consequence of the underlying processes, that surely obey energy conservation.
- Line 378-381: The formulation "In other words . . . " is indicating that the subsequent claim that "the decrease in water vapor over land as dictated by the Clausius Clapeyron equation . . . is muted by the decrease in latent heat flux" could be concluded from the foregoing remarks. But to do so one had in addition to know how strongly humidity changes per Kelvin in the simulations in order to compare with the Clausius Clapeyron prediction of 7%/K. But such a number is missing in the text.
- Line 420: What is the abbreviation "SOA" standing for?
- Lines 479-480: I do not see how from the spatial correlations between the total change in the energy balance and the individual $LW_{clear}$ term it can be concluded that the changes in the $LW_{clear}$ term is "largely a feedback" to the surface warming? I think that the strong correlations indicate only that the changes are a reaction to the warming. That a feedback on surface temperature is involved leading to mutual adjustment of temperature and the size of the $LW_{clear}$ term seems plausible independent of the value of the correlations, but the correlations give no indication on the importance of the feedbacks in determining the size of the $LW_{clear}$ term.
- Line 533: Typo: "biogeophysCIal".
- Lines 536-538: I think that this conclusion is wrong: From the mere fact that the 1%CO2 simulations from which TCR is determined contain the processes of the 1%CO2-bgc simulations from which the biogeophysical warming induced by $CO_2$ fertilization is derived, one cannot even expect that TCR and biogeophysical warming are positively correlated. Reason: First, the temperature rise in the 1%CO2 simulations is only slightly larger than in the radiatively coupled simulations 1%CO2-rad (where

biogeophysical warming from $CO_2$ fertilization is absent) (see e.g. Arora et al., 2020), meaning that TCR is mostly determined by the radiative effect of $CO_2$. But the biogeophysical warming from $CO_2$ fertilization arises from very different processes, mostly (as explained in the paper) from transpiration and albedo changes. So I do not see why a correlation between the two quantities should be "not necessarily unexpected". I think such a correlation is rather unexpected.

- Lines 538-539: In continuation of the previous comment: The remark that "the causes of larger TCR ... also operate in the 1PCTCO2-bgc runs" is only partially true, as the dominant effect determining TCR, namely the radiative $CO_2$ effect, is missing in the 1%CO2-bgc simulation. Hence, this remark also doesn't help to understanf why a positive correlation is found between TCR and biogeophysical warming from $CO_2$ ferilization.

- Lines 543-544: In continuation of the previous comment: The mere fact of a statistical correlation between TCR and biogeophysical warming from $CO_2$ ferilization (as exemplified by the mentioned three models) doesn't mean that this warming and TCR are "related" – where I suspect that "related" is meant here in a causal sense.

- Lines 544-546: In continuation of the previous comment: From the foregoing considerations based on correlations one can thus not conclude "the importance of the biogeophysical warming associated with carbon fertilization effect to intermodel variation of TCR". Nevertheless, I would surely agree that the considered biogeophysical warming contributes to TCR (and its intermodel variation as discussed in lines 547-555). But this can be concluded without reference to the diagnosed correlations (which, having value 0.55 (see line 531), is anyway rather weak). Hence, because the dominant mechanisms determining the considered biogeophysical warming and TCR are different, I think that this correlation is incidental, even though it is reported to be significant. Accordingly, in my opinion, this positive correlation should at most be mentioned as a curiosity.

- Lines 600-603: As explaind above, from the diagnosed statistical correlation one cannot conclude that "the causes of intermodel TCR variation ... are also responsible for some of the intermodel spread in the biogeophysical temperature response under carbon fertilization."

- Supplement Line 390: Typo: correLAtions instead of correALtions.

- All figures showing spatial distributions: As is often done, the author sets a dot into grid cells where the result is significant. Thereby the color impression gets for significant grid cells darker and a comparison with the color in the color bar gets biased. A better practice would be to set dots where results are not significant and a comparison with the color bar is anyway futile.

**References**

[1] Boucher, O., Jones, A., & Betts, R. A. (2009). Climate response to the physiological impact of carbon dioxide on plants in the Met Office Unified Model HadCM3. Climate

Dynamics, 32, 237-249.

[2] Cao, L., Bala, G., Caldeira, K., Nemani, R., & Ban-Weiss, G. (2009). Climate response to physiological forcing of carbon dioxide simulated by the coupled Community Atmosphere Model (CAM3.1) and Community Land Model (CLM3.0). Geophysical Research Letters, 36(10).

---

## Author Comment (AC1)

**Response to Reviewer #1**

I thank Reviewer #1 for their comments and for their time in evaluating the paper. My responses to each comment are below in blue.

In simulations where the radiative effect of rising $CO_2$ is switched off, nevertheless a warming is observed. This warming is a consequence of the reaction of the biogeochemical reaction of the land biosphere on rising $CO_2$, namely the physiological effect of $CO_2$ fertilization. This goes along with a closure of stomata and thus reduced transpiration, as well as an increased plant productivity typically leading to an increased growth and thus a reduction in albedo. These effects – reduced transpiration and reduced albedo – together with their consequences (e.g. changed cloudiness) explain the warming observed in those simulations as is known at least since the pioneering study by Bala et al., 2006, soon followed by investigations with other models (e.g. [1, 2] and Cao et al., 2010). The new study by Allen re-investigates this behaviour taking advantage of the recent CMIP6 multi-model ensemble. I don't have a complete overview on the literature on the subject, but I suspect that the new aspects of this study are the comparative investigation of results from a rather large ensemble of models, and methodologically the usage of an energy balance decomposition as main tool.

Yes, this is correct. I also now include a new analysis based on comments from Reviewer #2.

Overall the study is well written, I like in particular the rather concise summary in the "Conclusions" section. Nevertheless, the mentioning of so many numbers in the text and the usage of so many abbreviations makes reading tedious. I personally would have preferred to leave numbers to tables or illustrative figures and restrict the text to qualitative discussions – I don't think that in this way the scientific content would deteriorate. Moreover, I am not really sure whether all the figures are really needed, in particular since from the many geographic details seen in the maps almost nothing is discussed. Maybe a visualization of the size of the different terms of the energy balance would be more illuminating than the many maps. But this is largely a matter of style to be decided by the author.

I've added a bar chart of the SEB analysis to the Supplement. But I retain the spatial maps in the main text. Some of the spatial details are now discussed in the revision (e.g., southwest US). I also use bar charts for the new analysis, based on quantifying the carbon fertilization effect using the 1PCTCO2 minus 1PCTCO2-rad simulations (now abbreviated as FULL-RAD).

As far as I see, the results of this new study only confirm the insights obtained in previous investigations. Nevertheless, I think that because of the large range of models included in the analysis the paper could get an important reference on the subject. But for this a thorough review of the results of previous studies on the subject and also a comparison with their results should be added. Otherwise I fully support the publication of this study.

Additional review of previous studies added.

Major comments

1. The abstract should be more concise: Why do we need another study on the biogeochemical

effects of carbon fertilization? What is specific about this study? I think one should also mention that an energy decomposition is used to disentangle the different effects on the surface energy budget.

The above quoted studies (which are now cited) are based on a single model. As mentioned in the abstract, I use up to 15 models in this analysis. I have added the use of the surface energy balance decomposition to understand the drivers of the surface temperature response. I have also added an additional analysis that compares the carbon fertilization effect under a preindustrial background climate (as initially done) and under a warmer (higher atmospheric $CO_2$) background climate.

2. The text in line 86ff leaves the impression that the author is the first to find biogeophysically induced global warming. But this is not the first study quantifying the climate change caused by the physiological effects of $CO_2$ on plants (see above). Results of these and similar studies should be reported in the introduction. Moreover, the literature references on the subject should be completed (two additional references are found below, but I guess one finds more if one follows the citations).

I have cited additional studies and I indicate that the biogeophysically induced warming found here is consistent with these prior studies.

3. Completely missing is a comparison of the results of this study with those obtained in earlier studies on the same topic.

Qualitatively, the results in this study are similar to earlier studies, as now acknowledged in the Introduction. It's probably not particularly useful to try to compare quantitatively, since the various studies are based on different levels of atmospheric $CO_2$ concentration, e.g., many are based on a doubling of $CO_2$, whereas here the focus is on a quadrupling of $CO_2$. There are other nuances as well (e.g., background climate), which are now quantified in the revision.

Nonetheless, I have added numbers from Zarakas et al. (2020).

3. Names for the energy balance terms are not explicitly introduced and moreover those names are not consistently used. E.g. the term from latent heat is called "LH SEB term" in the text, "Latent Heat" in the titles of Fig. 3 and Supplementary Fig. 8, and "LH" in Supplementary tables 4 and 5. Because most of the paper centers around the discussion of these terms, I suggest that names for the different terms are explicitly introduced early in the paper, maybe already when the energy balance equation is introduced in section 2.2.

I have fixed all SEB acronyms according to the reviewer's suggestion. These are now clearly defined early in the paper, in Section 2.2.

4. Today all data sets have a unique identification by a DOI. Such DOIs should be provided for all data sets used.

Data set DOIs have been added to the Supplement.

5. The four paragraphs in lines 473-529 are discussing exclusively material from the

Supplement. In order that the main text is self-contained this material should either be included in the main text or that whole text passage should be moved to the Supplement.

I have moved much of this information to the Supplement.

Minor comments

First sentence of abstract is inappropriate here: Here it is stated that $CO_2$ fertilization is a "significant source of uncertainty in future climate projections". This is indeed correct, but the sentence leaves the impression as if the study could contribute to a reduction of this uncertainty. But this is not the case, so that the study cannot be motivated in this way. ⌞SEP⌟

Removed sentence.

Line 49: Preposition missing: ". . . associated with . . . " or ". . . associated to the . . . ". But even better would be to name also the direction of causality, e.g. "...increasing atmospheric $CO_2$ concentrations lead to carbon fertilization . . . ". ⌞SEP⌟

Fixed.

Line 64: I find it a bit misleading to denote the cooling induced by enhanced carbon storage as a "biogeochemical" effect. I would say that the cooling is caused physically (its a radiative effect) even though its induced biogeochemically. ⌞SEP⌟

Reworded according to the reviewer's suggestion.

Line 85: What does "Climate effects include ...as well as the drivers" mean? What "drivers" are "climate effects"? ⌞SEP⌟

Deleted "as well as the drivers".

Lines 137-138: Something is wrong with the grammar. ⌞SEP⌟

Missing word "of" added.

Line 138: The word "including" is superfluous since all eight models are explicitly ⌞SEP⌟named. ⌞SEP⌟

Deleted "including".

Lines 206-208: What means "not clear model differences"? It seems that the author ⌞SEP⌟expects that the dynamic vegetation modules have a particular impact on the changes in LAI, or what is the reason to consider here in particular the two models with dynamic vegetation?

Deleted sentence.

Line 304: The wording "more efficient stomata" is a bit weird, more efficient are not ⸢SEP⸥the stomata but is the water usage. ⸢SEP⸥

Fixed.

Lines 317-320 (similarly lines 344 and 488): I am not sure what exactly the reported ⸢SEP⸥"results . . . across models" are. I can imagine that the author has analyzed the correlation between transpiration and vaporation for each model separately, but if so I had expected that the author reports a mean correlation value plus a range indicating the spread across the different models, but in the text one finds only single numbers for the global and tropical regions. ⸢SEP⸥

Additional information under methods has been added.

There are two types of correlations ($r$) used throughout this manuscript. One is a spatial correlation between multimodel mean responses. Here, the multimodel mean responses at each grid box are first calculated and then a regional (e.g., global/tropical/extratropical) correlation is estimated. The second type is the intermodel correlation between model mean responses. Here, each model's grid box or regional (e.g., tropical) response is first estimated and then a correlation across models for that grid box (or region) is calculated. Significance of correlations is estimated from a two-tailed t-test as: $t = \dfrac{r}{\sqrt{\frac{1-r^2}{N-2}}}$, with $N$-$2$ degrees of freedom. N is either the number of grid boxes (for a spatial correlation) or the number of models (for correlations across models).

I have also added additional information specific to this section:

In addition, the increased evaporation is consistent with increased canopy interception from the greater LAI and subsequent evaporation from the canopy. For example, canopy evaporation increases by $1.02 \pm 0.92$ W m$^{-2}$ (9 of 13 models).

Line 334: The remark that increases occur "also" over land is only understandable when consulting te respective figure in the Supplement: Indeed from that figure it is visible that significant changes are mostly found over oceanic regions, which explains the "also". But the main text should stand for its own without such a consultation. ⸢SEP⸥

I've added the multimodel mean tropospheric specific humidity response over land only to the main text.

Lines 337-342: I cannot really follow this passage: What is called here the "former case" doesn't look for me like a "feedback" via the atmosphere but simply as a consequence of the changed temperature gradient due to the surface warming. And concerning the "latter case", the author seems to think that the mentioned "compensation" would be a process different from the mechanism described in the "former case", but I doubt that this can be justified by physics: "compensation" is not really a physical process but a consequence of the underlying processes, that surely obey energy conservation. ⸢SEP⸥

Reworded this paragraph.

Line 378-381: The formulation "In other words . . . " is indicating that the subsequent claim that "the decrease in water vapor over land as dictated by the Clausius Clapeyron equation . . . is muted by the decrease in latent heat flux" could be concluded from the foregoing remarks. But to do so one had in addition to know how strongly humidity changes per Kelvin in the simulations in order to compare with the Clausius Clapeyron prediction of 7%/K. But such a number is missing in the text.

This text has been deleted.

Line 420: What is the abbreviation "SOA" standing for?

Acronym defined, secondary organic aerosol.

Lines 479-480: I do not see how from the spatial correlations between the total change in the energy balance and the individual $LW_{clear}$ term it can be concluded that the changes in the $LW_{clear}$ term is "largely a feedback" to the surface warming? I think that the strong correlations indicate only that the changes are a reaction to the warming. That a feedback on surface temperature is involved leading to mutual adjustment of temperature and the size of the $LW_{clear}$ term seems plausible independent of the value of the correlations, but the correlations give no indication on the importance of the feedbacks in determining the size of the $LW_{clear}$ term.

Sentence deleted.

Line 533: Typo: "biogeophysCIal". Fixed.

Lines 536-538: I think that this conclusion is wrong: From the mere fact that the 1%CO2 simulations from which TCR is determined contain the processes of the 1%CO2-bgc simulations from which the biogeophysical warming induced by $CO_2$ fertilization is derived, one cannot even expect that TCR and biogeophysical warming are positively correlated. Reason: First, the temperature rise in the 1%CO2 simulations is only slightly larger than in the radiatively coupled simulations 1%CO2-rad (where biogeophysical warming from $CO_2$ fertilization is absent) (see e.g. Arora et al., 2020), meaning that TCR is mostly determined by the radiative effect of $CO_2$. But the biogeophysical warming from $CO_2$ fertilization arises from very different processes, mostly (as explained in the paper) from transpiration and albedo changes. So I do not see why a correlation between the two quantities should be "not necessarily unexpected". I think such a correlation is rather unexpected.

I have removed "not necessarily unexpected".

I do note, however, that these results are consistent with the prior work of Zarakas et al. (2020), which is now noted and cited in the revision.

Lines 538-539: In continuation of the previous comment: The remark that "the causes of larger TCR ...also operate in the 1PCTCO2-bgc runs" is only partially true, as the dominant effect determining TCR, namely the radiative $CO_2$ effect, is missing in the 1%CO2-bgc simulation. Hence, this remark also doesn't help to understanf why a positive correlation is found between TCR and biogeophysical warming from $CO_2$ ferilization.

I've deleted this sentence.

Lines 543-544: In continuation of the previous comment: The mere fact of a statistical correlation between TCR and biogeophysical warming from $CO_2$ ferilization (as exemplified by the mentioned three models) doesn't mean that this warming and TCR are "related" – where I suspect that "related" is meant here in a causal sense.

Deleted.

Lines 544-546: In continuation of the previous comment: From the foregoing consider- ations based on correlations one can thus not conclude "the importance of the biogeo- physical warming associated with carbon fertilization effect to intermodel variation of TCR". Nevertheless, I would surely agree that the considered biogeophysical warming contributes to TCR (and its intermodel variation as discussed in lines 547-555). But this can be concluded without reference to the diagnosed correlations (which, having value 0.55 (see line 531), is anyway rather weak). Hence, because the dominant mech- anisms determining the considered biogeophysical warming and TCR are different, I think that this correlation is incidental, even though it is reported to be significant. Accordingly, in my opinion, this positive correlation should at most be mentioned as a curiosity.

I've toned this down.

Lines 600-603: As explain above, from the diagnosed statistical correlation one cannot conclude that "the causes of intermodel TCR variation . . . are also responsible for some of the intermodel spread in the biogeophysical temperature response under carbon fertilization."

I've toned this down.

Supplement Line 390: Typo: correLAtions instead of correALtions.

Fixed.

All figures showing spatial distributions: As is often done, the author sets a dot into grid cells where the result is significant. Thereby the color impression gets for significant grid cells darker and a comparison with the color in the color bar gets biased. A better practice would be to set dots where results are not significant and a comparison with the color bar is anyway futile.

Thank you for the suggestion, but I have retained the current style.

References

[1] Boucher, O., Jones, A., & Betts, R. A. (2009). Climate response to the physiological impact of carbon dioxide on plants in the Met Office Unified Model HadCM3. Climate Dynamics, 32, 237-249.

[2] Cao, L., Bala, G., Caldeira, K., Nemani, R., & Ban-Weiss, G. (2009). Climate response to physiological forcing of carbon dioxide simulated by the coupled Com- munity Atmosphere Model (CAM3.1) and Community Land Model (CLM3.0). Geo- physical Research Letters, 36(10).

References added to the revision.

---

## Author Comment (AC2)

**Response to Reviewer #2**

I thank Reviewer #2 for their comments, in particular General Comment #1. This additional analysis has improved the scope of the paper. My responses to each comment are below in blue.

Review of *The Biophysical Effects of Carbon Fertilization of the Terrestrial Biosphere*

The paper examines climate system responses to the biophysical effects of carbon fertilization using a subset of the C4MIP experiments from CMIP6. The paper advances understanding by employing a surface energy balance decomposition, which allows the author to quantify the individual contributions of biosphere-induced climate system changes on surface temperature responses. The paper is well written, and the analyses are appropriate. I have several suggestions that I hope the author will address in the next draft.

**General Comments:**

1. The experiments here quantify the carbon fertilization effect by subtracting a Preindustrial climate from the end of the 1pctCO2 climate. However, you can also quantify the carbon fertilization effect by subtracting the end of the 1pctCO2-rad climate from the end of the 1pctCO2 climate. Calculating the CO2 fertilization effect using this method provides insight into the influence of the carbon fertilization effect within the context of a warmer (higher atmospheric CO2 concentration) climate. I suggest the author examine whether the climate system responses shown in the current manuscript are consistent with the climate system responses using this alternative method. They do not need to replicate all of the analyses, but should at least focus on replicating the results in the Main Figures 1-3. This will help to understand how robust the changes are to the background climate state, and may lead to new insights that inform the previous analyses.

Thank you for this suggestion. Although this has involved considerable additional work, I have performed the companion analysis using 1pctCO2 minus 1pctCO2-rad. Additional figures and an additional section are included in the revision.

2. There has been a considerable amount of work dedicated to the analysis of climate responses in these C4MIP-type experiments. The author should expand upon their description of these works. Indeed, much of the results presented here are supported by previous investigations. Some possible papers to describe (among many others):
   o Swann, A. L. S., F. M. Hoffman, C. D. Koven, and J. T. Randerson, 2016: Plant responses to increasing CO2 reduce estimates of climate impacts on drought severity. *Proc. Natl. Acad. Sci. USA*, *113*, 10 019–10 024, https://doi.org/10.1073/pnas.1604581113.
   o Skinner, C. B., C. J. Poulsen, and J. S. Mankin, 2018: Amplification of heat extremes by plant CO2 physiological forcing. *Nat. Commun.*, *9*, 1094, https://doi.org/10.1038/s41467-018-03472-w.
   o Lemordant, L., P. Gentine, A. S. Swann, B. I. Cook, and J. Scheff, 2018: Critical impact of vegetation physiology on the continental hydrologic cycle in response

to increasing CO2. *Proc. Natl. Acad. Sci. USA*, *115*, 4093–4098, https://doi.org/10.1073/pnas.1720712115.

o Zarakas, C. M., A. L. S. Swann, M. M. Laguë, K. C. Armour, and J. T. Randerson, 2020: Plant Physiology Increases the Magnitude and Spread of the Transient Climate Response to CO2 in CMIP6 Earth System Models. *J. Climate*, **33**, 8561–8578, https://doi.org/10.1175/JCLI-D-20-0078.1.

These references (and others) have been added to the revision.

3. The paper largely groups responses into the tropics vs extratropics. This is mostly appropriate given the distinct responses between those two latitude bands. However, there are some very interesting regional changes that the author should discuss. Namely, the zonally anomalous changes in the Southwest U.S. and parts of Western and Central Asia (e.g., increases in latent heating (from transpiration); reduced downwelling shortwave). These water-limited regions behave differently than energy-limited regions.

I have added some discussion on these water-limited regions.

4. Line 320-321: The author mentions that in areas with reduced transpiration, evaporation increases to satisfy the evaporative demand of the atmosphere. This may be partially true, but the primary reason for the increase in evaporation is likely because of increased canopy interception (from greater leaf area) and subsequent evaporation from the canopy.

I've added this point. Canopy evaporation increases by $1.02 \pm 0.92$ Wm$^{-2}$ (9 of 13 models).

5. Lines 385-388: The large contribution of the surface albedo change to warming in semi-arid regions is likely because these are water-limited areas that see large percentage increases in LAI. These are the regions we expect to see the greatest parentage increases in photosynthesis/fertilization

o Donohue, R. J., Roderick, M. L., McVicar, T. R. & Farquhar, G. D. Impact of CO2 fertilization on maximum foliage cover across the globe's warm, arid environments. *Geophys. Res. Lett.* **40**, 3031–3035 (2013).

Point made and reference added to the revision. The percentage change in LAI is relatively large in semi-arid regions, including for example the SW US (figure below).

[Figure]

6. Lines 390-392: You mention that the surface albedo effect is largest in the high latitudes due to the presence of snow and ice. Are the vegetation changes occurring in the presence of snow and ice? Aren't the vegetation responses to CO2 fertilization largely confined to the warmer season?

Yes, the annual mean albedo SEB term yields global land warming of $0.11 \pm 0.06$ K. In the tropics, the corresponding warming is smaller at $0.06 \pm 0.04$ K (11/15 models agree on the warming). In the extratropics, the corresponding warming is largest at $0.16 \pm 0.09$ K (13/15 models agree on warming). Based on the above figure showing the percent change in LAI, the larger extratropical warming due to surface darkening is not related to a larger percent increase in LAI. It is, however, consistent with the notion that vegetation induced surface darkening is larger when the vegetation covers brighter underlying surfaces, such as snow (e.g., Betts and Ball, 1997). There is obviously a seasonal dependency to this effect. For example, Li et al (2015) showed boreal forests have strong warming in winter and moderate cooling in summer with net warming annually, due to albedo effects dominating in winter and evapotranspiration effects dominating in summer, but with the albedo effects dominating in the annual mean.

Re-estimating the albedo SEB term by season in the extratropics (which is dominated by the Northern Hemisphere) shows the dominant warming effect occurs in DJF and MAM at $0.26 \pm 0.15$ K and $0.21 \pm 0.12$ K, respectively. Smaller warming occurs during JJA and SON at $0.07 \pm 0.05$ K and $0.11 \pm 0.06$ K, respectively. The larger SEB albedo term warming during the Northern Hemisphere cold months is consistent with the co-occurrence of snow and vegetation.

Betts, A. K., and J. H. Ball (1997), Albedo over the boreal forest, *J. Geophys. Res.*, 102(D24), 28901–28909, doi:10.1029/96JD03876.

Li, Y., Zhao, M., Motesharrei, S. *et al.* Local cooling and warming effects of forests based on satellite observations. *Nat Commun* **6**, 6603 (2015). https://doi.org/10.1038/ncomms7603

7. Can you speculate as to why GISS-E2-1-G simulates a decrease in LAI in response to CO2 fertilization? Is there something specific to the treatment of photosynthesis?

The GISS-E2-1-G vegetation model, the Ent Terrestrial Biosphere Model (Ent TBM; Schmidt et al., 2014; Kim et al., 2019; Ito et al. 2020) consists of multilayer canopy radiative transfer model (Spitters, 1986) and leaf gas exchange using the Ball-Berry stomatal conductance model (Ball and Berry, 1985) coupled with the Farquhar-von Caemmer photosynthesis model (Farquhar and von Caemmerer, 1982). Autotrophic and heterotrophic respiration is parameterized as in Kim et al. (2019).

GISS-E2-1-G prescribes fixed 2004 monthly LAI, so the model does not capture the impact of carbon fertilization on LAI. The change in LAI in GISS-E2-1-G is essentially zero at -0.00065. This has been clarified.

Ball, J. T., & Berry, J. A. (1987). A model predicting stomatal conductance and its contribution to photosynthesis under different environmental conditions. In I. Biggins (Ed.), Progress in Photosynthesis Research (Vol. IV, pp. 110–112). Dordrecht, Netherlands: Nijhoff.

Farquhar, G. D., & von Caemmerer, S. (1982). Modelling photosynthetic response to environmental conditions. In O. L. Lange, C. B. Osmond H. Ziegler (Eds.), Berlin, Encyclopedia of Plant Physiology (NS) (12B. pp. 549–587). Berlin: Springer.

Ito, G., Romanou, A., Kiang, N. Y., Faluvegi, G., Aleinov, I., Ruedy, R., et al. (2020). Global carbon cycle and climate feedbacks in the NASA GISS ModelE2.1. *Journal of Advances in Modeling Earth Systems*, 12, e2019MS002030. https://doi.org/10.1029/2019MS002030

Kim, D., Lee, M., & Seo, E. (2019). Improvement of soil respiration parameterization in a dynamic global vegetation model and its impact on the simulation of terrestrial carbon fluxes. *Journal of Climate*, 32(1), 127–143. https://doi.org/10.1175/JCLI-D-18-0018.1

Schmidt, G. A., Kelley, M., Nazarenko, L. et al. (2014). Configuration and assessment of the GISS ModelE2 contributions to the CMIP5 archive. *Journal of Advances in Modeling Earth Systems*, 6(1), 141–184. https://doi.org/10.1002/2013MS000265

Spitters, C. J. T. (1986). Separating the diffuse and direct component of global radiation and its implications for modeling canopy photosynthesis. Part II. Calculation of canopy photosynthesis. *Agricultural and Forest Meteorology*, 38(1-3), 231–242. https://doi.org/10.1016/0168-1923(86)90061-4

Technical Corrections:

1. Line 230-231: "Similar but less significant statements…". Reword this. It is unclear what less significant means here.

   Deleted "but less significant statements".